# The Expanding Role of Ketogenic Diets in Adult Neurological Disorders

**DOI:** 10.3390/brainsci8080148

**Published:** 2018-08-08

**Authors:** Tanya J. W. McDonald, Mackenzie C. Cervenka

**Affiliations:** Department of Neurology, Johns Hopkins University School of Medicine, 600 North Wolfe Street, Meyer 2-147, Baltimore, MD 21287, USA; twill145@jhmi.edu

**Keywords:** modified Atkins diet, epilepsy, glioblastoma multiforme, malignant glioma, Alzheimer’s disease

## Abstract

The current review highlights the evidence supporting the use of ketogenic diet therapies in the management of adult epilepsy, adult malignant glioma and Alzheimer’s disease. An overview of the scientific literature, both preclinical and clinical, in each area is presented and management strategies for addressing adverse effects and compliance are discussed.

## 1. Introduction

The ketogenic diet (KD) was formally introduced into practice in the 1920s although the origins of ketogenic medicine may date back to ancient Greece [1]. This high-fat, low-carbohydrate diet induces ketone body production in the liver through fat metabolism with the goal of mimicking a starvation state without depriving the body of necessary calories to sustain growth and development [2,3]. The ketone bodies acetoacetate and β-hydroxybutyrate then enter the bloodstream and are taken up by organs including the brain where they are further metabolized in mitochondria to generate energy for cells within the nervous system. The ketone body acetone, produced by spontaneous decarboxylation of acetoacetate, is rapidly eliminated through the lungs and urine. The classic KD is typically composed of a macronutrient ratio of 4:1 (4 g of fat to every 1 g of protein plus carbohydrates combined), thus shifting the predominant caloric source from carbohydrate to fat. Lower ratios of 3:1, 2:1, or 1:1 (referred to as a modified ketogenic diet) can be used depending on age, individual tolerability, level of ketosis and protein requirements [4]. To increase flexibility and palatability, more ‘relaxed’ variants have been developed, including the modified Atkins diet (MAD), the low glycemic index treatment (LGIT) and the ketogenic diet combined with medium chain triglyceride oil (MCT). Introduced in 2003, the MAD typically employs a net 10–20 g/day carbohydrate limit which is roughly equivalent to a ratio of 1–2:1 of fat to protein plus carbohydrates [5,6]. The LGIT recommends 40–60 g daily of carbohydrates with the selection of foods with glycemic indices <50 and ~60% of dietary energy derived from fat and 20–30% from protein [7]. The primary goal of this diet, primarily used in children, is not to induce metabolic ketosis and will not be further explored in this review. The MCT variant KD uses medium-chain fatty acids provided in coconut and/or palm kernel oil as a diet supplement and allows for greater carbohydrate and protein intake than even a lower-ratio classic KD [8], which can improve compliance. While there is an extensive literature documenting the use of KDs for weight loss and epilepsy [9,10], these diets have garnered increased interest as potential treatments of other diet-sensitive neurological disorders. The aim of the current review is to describe the evidence, preclinical and clinical, supporting KD use in the management of adult epilepsy, adult malignant gliomas and Alzheimer’s disease. Several randomized controlled trials support the use of KDs for the treatment of drug-resistant epilepsy and there is emerging evidence that these diets may also be effective in treating refractory status epilepticus, malignant glioma and Alzheimer’s disease in adults.

## 2. KDs in the Management of Adult Epilepsy and Refractory Seizures

Despite being first recognized as an effective tool in the treatment of epilepsy in the 1920s [11,12], interest in diet therapy subsequently waned following the introduction of anti-epileptic drugs (AEDs) until the 1990s. Studies and clinical trials emerged demonstrating its efficacy in patients with drug-resistant epilepsy and particular pediatric epilepsy syndromes [11,12,13]. In the management of drug-resistant epilepsy (seizures resistant to two or more appropriate AEDs), adult patients have a less than 5% chance of seizure freedom with additional drugs added and may not be surgical candidates due to a generalized epilepsy, multifocal nature, or non-resectable seizure focus [14,15]. Seizures that evolve into status epilepticus (prolonged seizure lasting longer than 5 minutes or recurrent seizures without return to baseline between seizures) despite appropriate first- and second-line AEDs are classified as refractory status epilepticus (RSE). If status epilepticus continues or recurs 24 h or more after the initiation of treatment with anesthetic agents to induce burst- or seizure-suppression, patients are diagnosed with super-refractory status epilepticus (SRSE) [16]. Growing preclinical and clinical evidence suggests that KDs can offer seizure reduction and seizure freedom in patients with drug-resistant epilepsy and status epilepticus through a variety of potential mechanisms.

There has been controversy over whether the major ketone bodies produced by the liver are responsible for the anti-seizure effect of the KD primarily due to the clinical observation that blood ketone (i.e., β-hydroxybutyrate) levels inconsistently correlate with seizure control amongst studies [17,18,19,20,21], although findings may relate to diet heterogeneity and methodological differences between studies. In addition, ketone levels at the neuronal or synaptic level may be a more accurate reflection of ketone effects on excitability [22] as opposed to systemic concentrations. As recently reviewed [23], an increasing number of compelling experimental studies highlight pleiotropic anti-seizure and neuroprotective actions of ketones. Such effects include ketone-induced changes in neurotransmitter balance and release as well as changes in neural membrane polarity to dampen the increased neuronal excitability associated with seizures. In rat models of epilepsy, acetoacetate and β-hydroxybutyrate increased the accumulation of γ-aminobutyric acid (GABA) in presynaptic vessels [24]. Ketotic rats, moreover, exhibit lower levels of glutamate in neurons but stable amounts of GABA, suggesting a shift in the total balance of neurotransmitters towards inhibition [25]. Supporting these pre-clinical data, humans maintained on a KD showed increased GABA levels in the cerebrospinal fluid and in brain using magnetic resonance spectroscopy [26,27]. Ketones can slow spontaneous neuronal firing in cultured mouse hippocampal neurons by opening adenosine tri-phosphate (ATP)-sensitive potassium channels [28,29]. Medium chain fatty acids, like decanoic acid, have also exhibited efficacy in *in vitro* and *in vivo* models of seizure activity. Decanoic acid application blocked seizure-like activity in hippocampal slices treated with pentetrazol and increased seizure thresholds in animal models of acute seizures using both the 6 Hz stimulation test (a model of drug-resistant seizures) and the maximal electroshock test (a model of tonic-clonic seizures), potentially through a mechanism involving selective inhibition of AMPA receptors [30,31,32].

Moreover, there may also be an additional neuroprotective benefit of ketogenic therapies related to improved mitochondrial function due to increased energy reserves combined with decreased production of reactive oxygen species (ROS) [33]. For example, the KD has been shown to stimulate mitochondrial biogenesis, increase cerebral ATP concentrations, and result in lower ROS production in animal models [34,35]. Animal models have similarly demonstrated that the KD may influence seizures associated with the mammalian target of rapamycin (mTOR) pathway, as rats fed a KD showed reduced insulin levels and reduced phosphorylation of Akt and S6, suggesting decreased mTOR activation and increased AMP-activated protein kinase signaling [36,37]. Ketone reduction of oxidative stress may occur via genomic effects, as ketone application in *in vitro* models inhibits histone deactylases (HDACs) resulting in increased transcriptional activity of peroxisome proliferator-activated receptor (PPAR) γ and upregulation of genes including the antioxidants catalase, mitochondrial superoxide dismutase and metallothionein 2 [38,39]. Lastly, there is emerging evidence that ketone bodies exhibit protective anti-inflammatory effects [40]. In animal models, KD treatment reduces microglial activation, expression of pro-inflammatory cytokines and pain and inflammation after thermal nociception [41,42,43]. Similar experimental work in non-epilepsy models suggested that ketone body anti-inflammatory effects may be mediated by hydroxy-carboxylic acid receptor 2 (HCA2) and/or inhibition of the innate immune sensor NOD-like receptor 3 (NLRP3) inflammasome [23,43,44]. These anti-inflammatory properties may explain the observed benefit of KD in treating patients with SRSE secondary to auto-immune and presumed auto-immune encephalitis such as those with new-onset refractory status epilepticus (NORSE) and febrile infection-related epilepsy syndrome (FIRES) [45].

In contrast to the aforementioned studies highlighting mechanisms mediated largely by ketones, recent preclinical work suggests the anti-seizure properties bestowed by KDs may instead relate to modulation of gut microbiota. The KD has been shown to alter the composition of gut microbiota in mice and ketosis is associated with altered gut microbiota in humans [46,47,48,49]. Studies using two mouse models of epilepsy (6 Hz stimulation test and mice harboring a null mutation in the alpha subunit of voltage-gated potassium channel Kv1.1) demonstrate that KD induced changes in gut microbiota, produced by feeding or fecal transplant, are necessary and sufficient to confer seizure protection. The effect appears to be mediated by select microbial interactions that reduce bacterial gamma-glutamylation activity, decrease peripheral gamma-glutamylated-amino acids and elevate bulk hippocampal GABA/glutamate ratios [50]. As rodent studies have shown different taxonomic shifts in response to KD therapy, the gut microbiota induced by KDs will depend on host genetics and baseline metabolic profiles [46,50]. Further research is needed to determine effects of the KD on microbiome profiles in adults with drug-resistant epilepsy and whether particular taxonomic changes in gut microbiota correlate with seizure severity and response to therapy.

A surge of clinical studies since the turn of the century support KD use in the management of chronic epilepsy in adults, with most reporting efficacy defined by the proportion of patients achieving ≥50% seizure reduction (defined as responders). A 2011 review pooled data from seven studies of the classic KD to show that 49% of 206 patients had ≥50% seizure reduction and, of these, 13% were seizure-free [51]. A 2015 meta-analysis reviewing ketogenic dietary treatments in adults from 12 studies of the classic KD, the MAD and the classic KD in combination with MCT found efficacy rates of KDs in drug-resistant epilepsy ranged from 13–70% with a combined efficacy rate of 52% for the classic KD and 34% for the MAD [52]. In the largest observational study of 101 adult patients naïve to diet therapy who subsequently started the MAD, 39% had ≥50% seizure reduction and 22% became seizure-free following 3 months of treatment [53]. Based on intention-to-treat (ITT) data from observational studies to date, the classic KD reduces seizures by ≥50% in 22–70% of patients while the MAD reduces seizures by ≥50% in 12–67% of patients [52,54,55], with some suggestion that dietary intervention may be more effective in patients with generalized rather than focal epilepsy [56,57].

Two randomized controlled trials (RCTs) evaluating MAD efficacy in adults with drug-resistant epilepsy have been reported recently. The first RCT in Iran compared the proportion of patients with focal or generalized epilepsy achieving ≥50% seizure reduction between 34 patients randomized to MAD use for 2 months (of whom 22 completed the study) compared to 32 control patients and found 35.5% (12/34) efficacy in the MAD group (ITT analysis) at 2 months compared to 0% in the control group [58]. These findings are in line with reports from meta-analyses of observational studies using MAD in adults [52]. The second RCT in Norway compared the change in seizure frequency following intervention in patients with drug-resistant (who had tried ≥3 AEDs) focal or multifocal epilepsy randomized to either 12 weeks of MAD (37 patients, of whom 28 received the intervention and 24 completed the study) or their habitual diet (38 patients, of whom 34 received the intervention and 32 completed the study) [59]. While they found no statistically significant difference in seizure frequency nor in 50% responder rate between the two groups following the intervention, a significant reduction in seizure frequency in the diet group compared to controls was observed among patients who completed the study but only for moderate benefit (25–50% seizure reduction). Importantly, compared to the patient population in the Iranian RCT with roughly half generalized and focal epilepsy patients (length of epilepsy 14–17 years on average, 6–9 mean seizures per month and had tried on average 3–4 AEDs), the Norwegian study investigated MAD treatment in adults with solely focal epilepsy who were particularly drug-resistant (length of epilepsy more than 20 years on average, with a median of 15 seizures per month and had tried on average 9–10 AEDs) and did note an improvement in overall seizure severity in the diet group, as measured by the Liverpool Seizure Severity Scale [58,59]. Additional RCTs of larger sample size are warranted to investigate MAD efficacy in different subpopulations of adult epilepsy patients.

Several case reports and case series have also demonstrated the successful use of KD therapy for management of RSE and SRSE [60,61,62,63,64]. For example, a case series of 10 adults with SRSE of median duration 21.5 days treated with a KD (either 4:1 or 3:1 ratio KD) showed successful cessation of status epilepticus in 100% of patients who achieved ketosis (9 out of 10 adults) at a median of 3 days (range 1–31 days) [65]. In the largest phase I/II clinical trial of 15 adult patients treated with a 4:1 ratio KD (14 of whom completed therapy) after a median of 10 days of SRSE, 11 (79% of patients who completed KD therapy, 73% of all patients enrolled) achieved resolution of seizures in a median of 5 days (range 0–10 days) [66]. As both RSE and SRSE carry high rates of morbidity and mortality [67], KDs offer a needed adjunctive strategy for management. KDs have the potential advantages of working rapidly and synergistically with other concurrent treatments; are relatively easy to start, monitor and maintain in the controlled intensive care unit setting with close follow up; do not contribute to hemodynamic instability seen with anesthetic agents and could potentially reduce the need for prolonged use of anesthetic drugs.

## 3. KDs in the Management of Adult Malignant Gliomas

Malignant gliomas are a highly heterogeneous tumor, refractory to treatment and the most frequently diagnosed primary brain tumor. Glioblastoma multiforme (GBM), the most aggressive type of glioma, carries an exceptionally poor prognosis with a median overall survival duration between 12 and 15 months from time of diagnosis and a 5-year survival rate of less than 5% [68,69]. The current standard of care for treating patients with GBM consists of maximal safe resection, followed by radiotherapy and concurrent chemotherapy with temozolomide [69]. Additional therapeutic strategies include glucocorticoid management of peritumoral edema and anti-angiogenic treatment with bevacicumab (Avastin); however therapeutic progress, particularly in regard to overall survival, remains poor [70]. Emerging research efforts over the past two decades seek to exploit a known cancer hallmark of abnormal energy metabolism in tumor cells named the “Warburg effect” following the discovery of physician, biochemist and Nobel laureate Otto Warburg that tumors exhibit high rates of aerobic glycolysis followed by predominant fermentation of pyruvate to lactate despite sufficient oxygen availability [71,72]. This metabolic phenotype confers several potential advantages to the cancer cell that include (1) more efficient generation of carbon equivalents for macromolecular synthesis; (2) bypassed mitochondrial oxidative metabolism and its concurrent production of reactive oxygen species and (3) acidification of the tumor site to facilitate invasion and progression [73]. As a result of this metabolic alteration, malignant glioma cells critically depend on glucose as the main energy source to survive and sustain their aggressive proliferative properties [74]. Moreover, clinical findings have identified hyperglycemia as a negative predictor of overall survival and a marker of poor prognosis in patients with GBM [75,76,77,78]. These findings have prompted nutritional strategies to target glycemic modulation using KDs, caloric restriction, intermittent fasting and combinatorial diet protocols broadly classified as ketogenic metabolic therapy.

Numerous preclinical studies have investigated KDs and/or exogenous supplementation of ketones or ketogenic agents in the treatment of malignant glioma. In the CT-2A malignant mouse astrocytoma model, a calorie-restricted KD decreased plasma glucose, plasma insulin-like growth factor and tumor weight when administered as a stand-alone therapy and elicited potent synergistic anti-cancer effects when administered in combination with glycolytic inhibitor 2-deoxy-d-glucose (2-DG) [79,80]. In the GL-261 malignant glioma model, KD fed mice had reduced peritumoral edema and tumor microvasculature, 20–30% increased median survival time and achieved complete and long-term remission when used concomitantly with radiation therapy [81,82,83]. A similar synergistic effect was observed between KD and temozolomide in the GL-261 model [84]. Comparable effects of KDs on tumor growth and survival time have also been shown in glioma derived mouse models of metastatic cancer and in patient-derived GBM subcutaneous and orthotopic implantation models [85,86]. These and other studies suggest that KDs induce a metabolic shift in malignant brain tissue towards a pro-apoptotic, anti-angiogenic, anti-invasive and anti-inflammatory state accompanied by a marked reduction in tumor growth in vivo [70] via mechanisms that include:(1)Reduction in blood glucose and insulin growth factor-1 levels [79];(2)Attenuated insulin activated Akt/mTOR and Ras/mitogen-activated protein kinase (MAPK) signaling pathways [87,88];(3)Induction of genes involved in oxidative stress protection and elimination of ROS through histone deactylase inhibition and altered expression of genes related to angiogenesis, vascular remodeling, invasion potential and the hypoxic response [38,82,84];(4)Enhanced cytotoxic T cell anti-tumor immunity [89]; and(5)Reduced inflammation via ketone body inhibition of the NLRP3 inflammasome and a reduction in other circulating inflammatory markers [43,90].

The first published case report in 2010 of an adult female patient with newly diagnosed GBM treated with a calorie restricted KD concomitant with standard care (radiation plus chemotherapy) following partial surgical resection demonstrated no tumor detection using fluorodeoxyglucose positron emission tomography (FDG-PET) and magnetic resonance imaging (MRI) after two months of treatment. However, after discontinuing diet therapy, tumor recurrence was detected 10 weeks later [91]. Subsequently a retrospective review reported 6 adult patients with newly diagnosed GBM treated with a KD, 4 of whom were alive at a median follow-up of 14 months and demonstrated reduced mean glucose compared to patients on a regular diet but only one patient was without evidence of disease for 12 months at the time of publication [92]. In another case report of 2 adult patients with recurrent GBM treated with a 3:1 calorie-restricted KD, both patients showed evidence of tumor progression by 12 weeks [93]. In the largest pilot trial, of 20 adult patients with recurrent GBM treated with a ≤60 g/day carbohydrate restricted diet, 3 discontinued because of poor tolerability, 3 had stable disease after 6 weeks that lasted 11–13 weeks and 1 had a minor response, with an overall trend towards an increase in progression-free survival in patients with stable ketosis [94]. A more recent case report documented an adult patient with newly diagnosed GBM who continued to experience significant tumor regression 24 months following combined treatment with subtotal resection, calorie restricted KD, hyperbaric oxygen and other targeted metabolic therapies [95]. These early observational, pilot studies and case reports principally provide evidence of feasibility and short-term safety as no serious adverse events were reported. Although they suggest a role for KDs in the management of GBM substantiated by an array of preclinical studies, given study design heterogeneity particularly in regard to diet formulation and calorie restriction, paucity of control groups and differences in endpoints, no conclusive statistical analysis of the clinical impact of KDs on adult GBM patient outcomes can be made. Consequently, a growing scientific interest has led to an increased number of registered clinical trials, including 4 randomized controlled trials (NCT02302235, NCT01754350, NCT01865162 and NCT03075514) of KDs (compared to a standard diet or between two KD types) in the management of adult GBM, 2 of which also include caloric restriction and a primary outcome of overall survival or progression-free survival [70,96].

## 4. KDs in the Management of Alzheimer’s Disease

In Alzheimer’s disease (AD), the most common form of progressive dementia, loss of recent memory and cognitive deficits are associated with extracellular deposition of amyloid-β peptide, intracellular tau protein neurofibrillary tangles and hippocampal neuronal death. Theories vary regarding the etiology of the overall disease process but mitochondrial dysfunction and glucose hypometabolism are recognized biochemical hallmarks [97]. Defects in mitochondrial function and a decline in respiratory chain function alter amyloid precursor protein (APP) processing to favor the production of the pathogenic amyloid-β fragment [98]. Reduced uptake and metabolism of glucose have been strongly linked to progressive cognitive degeneration, as neurons starve due to inefficient glycolysis [99]. Moreover, FDG-PET studies find asymptomatic individuals with genetic risk for AD or a positive family history show less prefrontal cortex, posterior cingulate, entorhinal cortex and hippocampal glucose uptake than normal-risk individuals. This reduction is associated with downregulation of the glucose transporter GLUT1 in the brain of individuals with AD [40,100]. Increasing evidence has demonstrated an association between high-glycemic diet and greater cerebral amyloid burden in humans [101] and that increased insulin resistance contributes to the development of sporadic AD [102,103], suggesting diet as a potential modifiable behavior to prevent cerebral amyloid accumulation and reduce AD risk.

Preclinical work supports the role of ketogenic therapies to prevent or ameliorate histological and biochemical changes related to Alzheimer’s disease pathology. *In vitro* studies showed attenuation of deleterious amyloid-β induced effects on rat cortical neurons by pre-treatment with coconut oil (containing high concentrations of MCT) or medium chain fatty acids via activation of Akt and extracellular-signal-regulated kinase (ERK) signaling pathways [104]. Similarly, preclinical studies using animal models of dementia demonstrated reduced brain amyloid-β levels, protection from amyloid-β toxicity and better mitochondrial function following administration of the KD, ketones and MCT [105,106,107,108]. Importantly, ketone body suppression of mitochondrial amyloid entry has been further shown to improve learning and memory ability in a symptomatic mouse model of AD [109]. In aged rats, a KD administered for 3 weeks improved learning and memory and was associated with increased angiogenesis and capillary density suggesting the KD may support cognition through improved vascular function [110]. In summary, these preclinical observations provide insight into potential mechanisms through which KDs and ketones may influence AD risk and pathology and lay the foundation for subsequent studies in humans.

In the first randomized controlled trial in humans, 20 patients with AD or mild cognitive impairment (MCI) received a single oral dose of either MCT or placebo on separate days and demonstrated expected elevations in serum ketone level following ingestion but only patients without the Apolipoprotein E (APOE) ε4 allele showed enhanced short-term cognitive performance on a brief screening tool measuring cognitive domains that included attention, memory, language and praxis [111]. This study was later replicated with similar improvements in working memory, visual attention and task switching seen in 19 elderly patients without dementia who received the MCT supplement [112]. Another RCT in adults with MCI treated with either a very low (5–10%) or high (50%) carbohydrate diet over 6 weeks showed an improvement in verbal memory performance that correlated with ketone levels in the ketogenic diet group [113]. A 2015 case report suggested that regular ketone monoester ((R)-3-hydroxybutyl (R)-3-hydroxybutyrate) supplementation, rather than a change to habitual diet, produced repeated diurnal elevations in circulating serum β-hydroxybutyrate levels and improved cognitive and daily activity performance over a 20-month period [114]. A single-arm pilot trial in 15 patients with mild-moderate AD using an MCT-supplemented ≥ 1:1 ratio KD for 3 months showed an improved Alzheimer’s disease Assessment Scale –cognitive subscale score in 9 out of 10 patients who completed the study and achieved ketosis (as measured by elevated serum β-hydroxybutyrate levels at follow-up) [115]. Three additional studies in patients with MCI or mild-moderate AD using at least 3-month treatment protocols (2 randomized studies of MCT or a ketogenic product compared to placebo for 3–6 months and 1 observational study administering a ketogenic meal over 3 months) reported that the cognitive benefit of ketogenic therapies was greatest in patients who did not have the APOE ε4 allele [116,117] and, in the observational study, was limited to APOE ε4 negative patients with mild AD [118]. A recent study of patients with mild-moderate AD treated with 1 month of MCT supplements demonstrated increased ketone consumption, quantified by brain ^11^C-acetoacetate PET imaging before and after administration, suggesting ketones from MCT can compensate for the brain glucose deficit observed in AD [119]. The clinical evidence lends support for the use of KDs and/or supplements to improve cognitive outcomes in patients with AD, however results indicate that stage/level of disease progression and APOE ε4 genotype may affect response to dietary treatment. Ongoing registered randomized clinical trials sponsored by Johns Hopkins University (NCT02521818), Wake Forest University (NCT03130036, NCT03472664 and NCT02984540), Université de Sherbrooke (NCT02709356) and the University of British Columbia (NCT02912936) are underway (active, recruiting, or completed) in individuals with subjective memory impairment, mild AD, and/or healthy controls to evaluate the impact of:(1)6–18 weeks of a modified Ketogenic-Mediterranean diet compared to a low-fat diet;(2)12 weeks of MAD compared to a recommended diet for seniors to achieve a healthy eating index;(3)1 month treatment with two different MCT oil emulsions (60–40 oil or C8 oil); or(4)10 days, twice a day, supplementation with a lactose-free skim milk drink containing either 10–50 g/day of MCT oil or 10–50 g/day of placebo (high-oleic sunflower oil)

On primary outcomes that include brain acetoacetate/glucose metabolism using PET, AD biomarkers, level of serum ketones, safety and feasibility as well as secondary outcomes that include cognition, function and examining key treatment response variables such as APOE genotype, amyloid positivity and metabolic status that could inform precision medicine approaches to dietary prescription.

## 5. Management of Adverse Effects and Poor Compliance in Adults

The most commonly reported adverse effects associated with KD use in adults with epilepsy and long-term diet use in children with epilepsy are gastrointestinal effects, weight loss and a transient increase in lipids [120,121]. Similar side effects have been reported in clinical studies of KD use in malignant glioma and AD, although a true assessment of risk in these populations is difficult due to the small number of trials, short duration of follow up and heterogeneity in KD therapy applied [70,122]. The gastrointestinal side effects which include constipation, diarrhea and occasional nausea and vomiting are typically mild, improve with time, can often be managed with diet adjustments with the guidance of a dietitian or nutritionist and infrequently require medical intervention. Smaller meals, increased fiber intake, exercise and increased sodium and fluid intake can often prevent or alleviate these complaints. Weight loss may be an intended positive effect in patients who are overweight but for those who want to maintain or gain weight, adjustments in caloric intake are recommended. This is of particular importance in patients with malignant glioma as the development of cachexia, due to weight loss principally affecting skeletal muscle mass, is associated with decreased cancer therapy tolerance and impaired respiratory function, leading to lower survival rates [123]. Increases in serum lipids have been shown to normalize with continued diet therapy (after 1 year) or return to normal after cessation of diet therapy in adult epilepsy patients [57,124,125]. In addition, very low carbohydrate diets that induce ketosis have been shown to lead to reductions in serum triglycerides, low-density lipoprotein and total cholesterol and increased levels of high-density lipoprotein cholesterol in adults [9]. Other potential side effects can result from vitamin and mineral deficiencies secondary to restricting carbohydrates and prolonged ketonemia, including osteopenia and osteoporosis [3,126,127], although the precise mechanism remains unclear. The standard practice of supplementing a recommended daily allowance of multivitamin and mineral supplements can reduce the risk of such deficiencies.

Diet adherence and compliance remain significant barriers to successful implementation and an adequate assessment of KD efficacy. Common methodologies to assess and document KD adherence in adults beyond patient self-report include frequent measurements of serum β-hydroxybutyrate or urine acetoacetate concentrations during the first few weeks on the diet and/or collection of dietary food records [57,121,128]. As examples, daily urine ketone assessments are traditionally used in adults with epilepsy during MAD initiation until moderate to large levels of ketosis are reached and serum ketone assessments using drops of blood from a finger stick have been used to guide short-term KD therapy in patients with GBM [53,128]. Still, the majority of studies traditionally report adherence based on patient self-report. A combined adherence rate of 45% for all KD types, 38% for the classic KD and 56% for the MAD, has been reported in a review of the epilepsy literature [52]. In the largest observational study of 139 adult epilepsy patients treated with KDs, 48% (67/139) discontinued the diet (39%) or were lost after initial follow up (9%) with approximately half of patients citing difficulty with compliance or restrictiveness as the reason for stopping [53]. The literature of adherence rates in adult GBM and AD is sparse but growing, with the largest GBM study reporting 15% (3/20) drop-out after 2–3 weeks due to subjectively decreased quality of life [94] and a recent 3 month single-arm AD pilot trial of a MCT-supplemented KD reporting 33% (5/15) attrition due to caregiver burden [115]. Often the provision of food recipes and resources to patients and families during initial diet training and subsequent visits can emphasize the variety of food choices and ease of use rather than perceived restrictiveness. Additional methods to improve adherence and compliance, as well as access for patients who live distant from a KD center, include scheduled telephone calls or electronic communication with the supervising dietitian or nutritionist, provision of ketogenic supplements and use of electronic applications like KetoDietCalculator^TM^ (The Charlie Foundation for Ketogenic Therapies, Santa Monica, CA, USA) to prevent drop-out and emphasize progress and success [6,129].

## 6. Conclusions

Although the neurological conditions discussed in this review-epilepsy, malignant glioma and Alzheimer’s disease—have distinct disease processes, each exhibit disrupted energy metabolism, increased oxidative stress and neuro-inflammation. As each of these pathophysiologic factors can be influenced through diet manipulation, it is logical and reasonable that diet could alter the course and outcomes of these and other neurologic disorders that share common pathways. Extensive preclinical work supports the use of KDs and/or ketone bodies to thwart or ameliorate histological and biochemical changes leading to neurologic dysfunction and disease. Demonstrated and hypothesized mechanisms by which ketogenic therapies influence epilepsy, malignant glioma and Alzheimer’s disease include metabolic regulation, neurotransmission modulation, reduced oxidative stress and anti-inflammatory and genomic effects that were highlighted in this review and summarized in Table 1. In some instances, an understanding of the mechanisms by which the KD and ketones exert their effects has led to novel therapeutic targets and work to develop new pharmaceutical drugs [130]. For many disorders, the clinical literature is still growing and limited by the conditions that make dietary interventions difficult to evaluate. For example, evaluation of whole diet changes cannot be performed under blinded circumstances as the participant will be aware of the diet changes made and/or the content of their meals. Other methodological constraints relate to limitations in inter-study comparison due to the heterogeneity of diet intervention used and reduced statistical power to detect significant effects when baseline levels of nutrient intake and individual variability are appropriately controlled. There are also challenges in monitoring diet compliance in the ambulatory setting as adherence to a prescribed diet can be more difficult to achieve than with a traditional pharmaceutical intervention. However, dietary interventions have the advantage of being non-invasive, relatively low risk and generally without serious adverse effects in the appropriate clinical context and may be particularly useful as an adjunctive therapy that synergizes with other pharmacologic and non-pharmacologic approaches. The scientific evidence collected from clinical studies in humans to date has supported KD therapy use in adult epilepsy, adult malignant glioma and Alzheimer’s disease, although overall assessment of efficacy remains limited due to study heterogeneity and indications that particular patient subpopulations may achieve disparate levels of benefit. Further clinical investigation using more standardized KD protocols and in patient subpopulations is warranted.

## Figures and Tables

**Table 1 brainsci-08-00148-t001:** Hypothesized mechanisms through which ketogenic therapies influence neurological disease.

Ketogenic Mechanisms	Epilepsy	Malignant Glioma	Alzheimer’s Disease
***Metabolic Regulation***			
↓Glucose uptake & glycolysis	+	+	
↓Insulin, IGF1 signaling		+	+
↑Ketones/ketone metabolism	+		+
Altered gut microbiota	+		
***Neurotransmission***			
Altered balance of excitatory/inhibitory neurotransmitters	+		
Inhibition of AMPA receptors	+		
↓mTOR activation & signaling	+	+	
Modulation of ATP-sensitive potassium channels	+		
***Oxidative Stress***			
↓Production of reactive oxygen species	+	+	
↑Mitochondrial biogenesis/function	+		+
***Inflammation/Neuroprotection***			
↓Inflammatory cytokines	+	+	
NLRP3 inflammasome inhibition	+	+	
↑cytotoxic T cell function		+	
↓peritumoral edema		+	
↓amyloid-β levels			+
***Genomic Effects***			
Inhibition of HDACs	+	+	
↑PPARγ	+		
↓Expression of angiogenic factors in tumor cells		+	

AMPA—α-amino-3-hydroxyl-5-methyl-4-isoxazolepropionic acid; IGF1—insulin-like growth factor 1; HDACs—histone deacetylases; mTOR—mammalian target of rapamycin; NLRP3—NOD-like receptor protein 3; PPAR—peroxisome proliferator-activated receptor. ↓—decreased; ↑—increased; +—mechanism shown in *in vitro* or *in vivo* studies.

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
