# Peer review of "The Expanding Role of Ketogenic Diets in Adult Neurological Disorders"

_brainsci, 2018, doi:10.3390/brainsci8080148_

Reviewer 1 Report
Well written review detailing the state of the art for ketogenic diets as possible therapy for seizures, primary aggressive brain malignancies like glioblastoma multiforme and alzheimer’s disease. The paper is a concise review that arrives at carefully stated conclusions about the state of the art using ketogenic diets for therapeutic intent. The authors are careful to state what is known and accepted from what might be investigated in the future.
Minor suggestions:
1. The few side effects recorded with the ketogenic diets is emphasized by a manuscript that documents the lack of long term problems in epileptic children treated with the diet for many years. The addition of this reference would help to strengthen the contention that the diet is safe. (1)
2. The authors correctly point out that compliance with a ketogenic diet, like any diet, can be difficult to monitor. Patients with diabetes are monitored for ketosis using drops of blood from a finger stick. This methodology has been used to guide ketogenic diets for short term therapy, 6 weeks, in patients with glioblastoma multiforme treated with a ketogenic diet along with standard of care radiation and chemotherapy, temozolomide. Monitoring blood ketone levels has the advantage of providing documentation that the patient has achieved and remained ketotic, or not, during the treatment. (2)
1. Patel A, Pyzik PL, Turner Z, Rubenstein JE, Kossoff EH. Long-term outcomes of children treated with the ketogenic diet in the past. Epilepsia. 2010;9999(9999).
2. Schwartz KA, Noel M, Nikolai M, Chang HT. Investigating the Ketogenic Diet As Treatment for Primary Aggressive Brain Cancer: Challenges and Lessons Learned. Frontiers in Nutrition. 2018;5(11).
Author Response
Reviewer #1:
Well written review detailing the state of the art for ketogenic diets as possible therapy for seizures, primary aggressive brain malignancies like glioblastoma multiforme and alzheimer’s disease. The paper is a concise review that arrives at carefully stated conclusions about the state of the art using ketogenic diets for therapeutic intent. The authors are careful to state what is known and accepted from what might be investigated in the future.
We appreciate the reviewers comment and have made revisions as noted below.
Minor suggestions:
1. The few side effects recorded with the ketogenic diets is emphasized by a manuscript that documents the lack of long term problems in epileptic children treated with the diet for many years. The addition of this reference would help to strengthen the contention that the diet is safe. (1)
We have added the suggested reference (#120) by the reviewer to line 326-328 as well as reference to an additional review article (#121) exploring diet tolerability in children and adults.
2. The authors correctly point out that compliance with a ketogenic diet, like any diet, can be difficult to monitor. Patients with diabetes are monitored for ketosis using drops of blood from a finger stick. This methodology has been used to guide ketogenic diets for short term therapy, 6 weeks, in patients with glioblastoma multiforme treated with a ketogenic diet along with standard of care radiation and chemotherapy, temozolomide. Monitoring blood ketone levels has the advantage of providing documentation that the patient has achieved and remained ketotic, or not, during the treatment. (2)
We have revised/ expanded lines 351-357 to include information on common methodologies of assessing and documenting adherence – measurement of serum and urine ketones – using the example and reference suggested by the reviewer (#128).
1. Patel A, Pyzik PL, Turner Z, Rubenstein JE, Kossoff EH. Long-term outcomes of children treated with the ketogenic diet in the past. Epilepsia. 2010;9999(9999).
2. Schwartz KA, Noel M, Nikolai M, Chang HT. Investigating the Ketogenic Diet As Treatment for Primary Aggressive Brain Cancer: Challenges and Lessons Learned. Frontiers in Nutrition. 2018;5(11).
Reviewer 2 Report
The authors have performed a study to investigate the possible therapeutic relevance of ketogenic diets (KDs) in adult neurological disorders. By means of several disease conditions such as epilepsy, glioma, and Alzheimer’s disease, the current review highlights the possibility that KD-feeding exerts protective effects mediated by multiple factors and diverse aspects. In general, the authors provide compelling experimental evidences to support their conclusion. Furthermore, the manuscript is well constructed and its main points are well articulated.
Minors:
Line 17: Except for the first sentence, please use the following abbreviation (Ketogenic diet à KD).
Line 35-36: The principle goal of this diet, primarily used in children, is not to induce metabolic ketosis and will not be further explored in this review.
Line 44: ----drug-resistant epilepsy, and
Line 48: After being à Despite being..
Line 175: The current standard of care for the treatment of patients à The current standard of care for treating patients with …
Line 229: tumor progression at by (please pick one) 12 weeks.
Line 245: primary outcome of à a primary outcome..
Line 257: … normal-risk individuals. This reduction is…
Line 298: observational study, was limited to
Line 350: … MAD, has been..
Line 390: .. Alzheimer’s disease, although..
Author Response
Reviewer #2
The authors have performed a study to investigate the possible therapeutic relevance of ketogenic diets (KDs) in adult neurological disorders. By means of several disease conditions such as epilepsy, glioma, and Alzheimer’s disease, the current review highlights the possibility that KD-feeding exerts protective effects mediated by multiple factors and diverse aspects. In general, the authors provide compelling experimental evidences to support their conclusion. Furthermore, the manuscript is well constructed and its main points are well articulated.
We appreciate the reviewers comment and have made revisions as noted below.
Minors:
Line 17: Except for the first sentence, please use the following abbreviation (Ketogenic diet à KD).
We have used the abbreviation KD in the place of ketogenic diet as suggested.
Line 35-36: The principle goal of this diet, primarily used in children, is not to induce metabolic ketosis and will not be further explored in this review.
We have revised the sentence as suggested.
Line 44: ----drug-resistant epilepsy, and
We have revised the sentence as suggested.
Line 48: After being à Despite being..
We have revised the sentence as suggested.
Line 175: The current standard of care for the treatment of patients à The current standard of care for treating patients with …
We have revised the sentence as suggested.
Line 229: tumor progression at by (please pick one) 12 weeks.
We have revised the sentence as suggested.
Line 245: primary outcome of à a primary outcome..
We have revised the sentence as suggested.
Line 257: … normal-risk individuals. This reduction is…
We have revised the sentence as suggested.
Line 298: observational study, was limited to
We have revised the sentence as suggested.
Line 350: … MAD, has been..
We have revised the sentence as suggested.
Line 390: .. Alzheimer’s disease, although..
We have revised the sentence as suggested.